# A Universal Biofilm Reactor Sensor for the Determination of Biochemical Oxygen Demand of Different Water Areas

**DOI:** 10.3390/molecules27155046

**Published:** 2022-08-08

**Authors:** Liang Wang, Huan Lv, Qian Yang, Yiliang Chen, Junjie Wei, Yiyuan Chen, Ci’en Peng, Changyu Liu, Xiaolong Xu, Jianbo Jia

**Affiliations:** School of Biotechnology and Health Sciences, Wuyi University, Jiangmen 529020, China

**Keywords:** biochemical oxygen demand, biofilm reactor sensor, domesticated microorganisms, universality, microbial community structure

## Abstract

In this study, we developed a simple strategy to prepare a biofilm reactor (BFR) sensor for the universal biochemical oxygen demand (BOD) determination. The microorganisms in fresh water were domesticated by artificial seawater with different salinity gradients successively to prepare the BFR sensor. The prepared BFR sensor exhibits an efficient ability to degrade a variety of organic substances. The linear range of BOD determination by the BFR sensor is 1.0–10.0 mg/L^−1^ with a correlation coefficient of 0.9951. The detection limit is 0.30 mg/L according to three times of signal-to-noise ratio. What is more, the BFR sensor displayed excellent performances for the BOD determination of different water samples, including both fresh water and seawater. The 16S-rRNA gene sequencing technology was used to analyze the microbial species before and after the domestication. The results show that it is a general approach for the rapid BOD determination in different water samples.

## 1. Introduction

Biochemical oxygen demand (BOD) is the amount of dissolved oxygen (DO) needed by aerobic biological organisms to break down organic material present in the given water sample [1]. It is one of the most important parameters for water quality evaluation. The conventional BOD method is the well-known BOD_5_ method, which takes 5 days of incubation at 20 °C in the dark [2]. Although BOD_5_ is a good indicator of the water quality, the biochemical oxidation is a slow process and the test needs long time to achieve the result. To monitor the water quality in time, the rapid BOD measuring methods are of great interest. Since the first BOD biosensor was reported by Karube et al. [3], the microbial biosensors were studied extensively because they can monitor BOD rapidly. The BOD biosensors are usually based on the selected microorganisms that were immobilized on the DO probe by measuring the respiration rate. For the application of BOD biosensors, the key issues are the selection of suitable microorganisms and corresponding immobilization methods [4,5,6]. The stability of the BOD biosensor was greatly affected by the immobilized method. Especially, the biodegradability of the microorganism that was immobilized on the BOD biosensor plays a critical role in the application scope of the water body [7,8]. Most of them choose a single strain, as they exhibit good performances for the standard solution. However, the universality is often poor for the practical water samples. The selective biodegradability of the immobilized microorganisms limits their practical use severely [8,9,10,11]. Only the sensors based on suitable microorganisms, which adapt to the environmental water body, can obtain valuable results in practical applications [12,13]. For example, microbial biosensors that can be used in the fresh water cannot be used in the high-salinity seawater [14,15]. It is reported that a high salinity environment significantly influences the microbial communities [16]. Additionally, elevated salinity has inhibitory or toxic effects on the microbial activity and population structure in bioreactors, and it would reduce organic carbon removal efficiency [17,18]. Only the microorganisms cultured from seawater can survival in high-salinity water, which are suitable for the degradation of organics in the seawater [15,19]. As far as we know, there is no universal microbial sensor that can be used for the rapid BOD determination of different kinds of water bodies.

Recently, we developed a new analytical method that embraces the advantages of both the BOD_5_ method (wide applicability) and rapid BOD biosensors (rapidity) [19,20,21,22,23,24,25]. Such a method employs a biofilm reactor (BFR) sensor that was cultivated with naturally microorganisms and can achieve rapid and indiscriminative biodegradability. The BFR system embraces many notable advantages, such as simplicity in device, convenience in operation, and minimal maintenance, making it promising for practical BOD real-time warnings. The BFR sensor can be used for online BOD determination for either fresh water or sea water, depending on the selection of domesticated water sources. It overcomes the limitations of the previous BOD biosensor and improves the stability and service life to a great extent. It can be effectively applied to the water quality monitoring of coastal areas with high salt content [19]. What is more, the online analyzer based on the BFR sensor can continually operate over 30 days without human intervention and additional chemical reagent consumption. It can track the fluctuation of the biodegradable organic compound level timely and accurately [25]. However, it still cannot be used as the universal BOD detector for both fresh water and sea water at the same time.

In this study, the natural microbial sources extracted from fresh water were domesticated directly by artificial seawater with different salinity gradients. The domesticated method is easy to operate, and the microbial survival rate is improved. The BFR sensor is universal in the rapid BOD detection of water samples in different regions. The stability and salinity tolerance of the reactor were tested, and the degradation ability of different organic compounds was analyzed. The BFR sensor can be used for rapid BOD measurement of fresh water and seawater at the same time. The results were comparable to that of the traditional BOD_5_. The 16S-rRNA gene sequencing technology was used to compare the microbial species before and after domestication.

## 2. Experimental

### 2.1. Material and Sample Preparation

The analytical reagent-grade glucose, glutamic acid, etc. were purchased from Macklin (Shanghai Macklin Biochemical Co., Ltd., Shanghai, China) and used to determine the BFR sensor performances. The chemicals used in this study are listed in Appendix A (List of Experimental Reagents) in the supporting information. The glucose and glutamic acid (GGA) solution (BOD_5_ = 198.0 mg O_2_ L^−1^) were prepared according to the standard method [2]. GGA, artificial seawater, and all other solutions were prepared with ultrapure water (Milli-Q) in all the experiments. Polylactic acid (PLA) powder with the purity of 90% was brought from Polymaker (Jufu science and technology, Shanghai, China). The Weiming pond water in Wuyi University was selected as the microbial source in this study, and the standard BOD_5_ method was used for water sample measurement at the same time.

### 2.2. Preparation of Biofilm Reactor

A porous cylindrical PLA pipe was produced by a homemade 3D printer with a length of 7.0 cm, an inner diameter of 2.5 cm, and an aperture of 2.5 mm, respectively, as given in Figure 1. The Weiming pond water of Wuyi University was selected as the domesticated microbial source. Firstly, the pond water extracted on site was continuously aerated and pumped to the PLA pipe at a flow rate of 3.0 mL/min and a constant temperature of 35 °C The microbial film was formed on the inner wall of the PLA pipe gradually after a period of culture. At this time, the initial cultivation of the microbial membrane can be used for the detection of fresh water. Next, in order to expand the universality of the BFR sensor for the quality monitoring of seawater, a gradient of salinity of artificial seawater was selected to domesticate the microbial film, which can be salt-tolerant. The prepared BFR was circulated with 0.20, 0.40, 0.70, and 1.00 mol/L NaCl artificial seawater from low to high concentration successively and cultured with each solution for 2–3 days, respectively. After the domestication, some inactivated microorganisms would fall off from the inner wall of the reactor, and the remaining surviving microorganisms would continue to reproduce and cultivate. The BFR sensor can be used for the rapid determination of BOD in both fresh water and seawater at the same time.

### 2.3. Measurement Procedure

A three-electrode DO probe was used. The working electrode was covered with a Teflon film (Orbisphere 2956A, Hach, Loveland, CO, USA). All current signals were measured under constant applied potential of −700 mV versus Ag/AgCl (0.10 M KCl) and controlled by an electrochemical workstation (CHI832B, Chenhua, Shanghai, China). The BFR sensor and the water sample were placed in a 35 °C constant temperature water bath. The fully aerated water sample passed through the DO probe directly at certain flow rate and output a stable current value. Then, the water sample passed through BFR and output a stable current value through the DO probe. The measurement procedure takes 20 min. The difference between the current values is the analysis signal.

## 3. Results and Discussion

### 3.1. Performance Optimization

#### 3.1.1. The Influence of Temperature

The microbial species in the extracted natural source are complex, and the adaptability of these microorganisms is also different. The test temperature will affect the microbial activity and the performance of the reactor obviously [14,26,27]. Figure 1 shows that the influence of temperature in the range of 20–40 °C on the BFR sensor activity. It can be seen that the activity of BFR sensor is not significantly affected under the investigated temperatures, indicating that the degradation ability of the domesticated natural microorganisms to the temperature is roughly stable. The response signal is the largest at 35 °C, therefore, it was selected as the optimal temperature in the following experiments.

#### 3.1.2. The Influence of Flow Rate

When the water sample is in contact with the microorganisms on the BFR inner surface, the microorganisms will degrade the organic matters in the sample. The sensitivity will be affected by the sample duration that the water sample stays on the BFR surface. The longer the reaction time is, the more organic matters will be degraded. However, it will prolong the measurement time and cannot reflect the water quality in time if the flow rate is too slow. What is more, the longer reaction time can lead to more DO consumption in the reactor. Therefore, according to various considerations, the influence of the flow rate of 1.0–5.0 mL/min on the BFR sensor response was tested. As shown in Figure 2, the current response increased within the flow rate range of 1.0–3.5 mL/min and decreased within the folw rate range of 3.5–5.0 mL/min. Therefore, 3.5 mL/min was selected as the best flow rate.

#### 3.1.3. Salt Tolerance of the BFR Sensor

The domesticated microorganisms from fresh water are different from the natural microorganisms. They have good adaptability to both fresh water and seawater. The seawater with high salinity will affect the permeability of freshwater microorganisms. Correspondingly, the biological activity and degradation ability of natural microorganisms were also changed significantly. Therefore, they cannot be used for the BOD measurement of seawater samples at the same time. However, after the domestication with artificial seawater of different salinity gradients, a large number of microorganisms that can adapt to high salinity were survived. The surviving microorganisms possess enough activity in both fresh water and seawater environment. Therefore, the BFR sensor can be used not only for the BOD determination of fresh water, but also for the BOD determination of seawater at the same time. It expanded the application scope of the BOD determination greatly. The signal responses of the domesticated microorganisms to 5.0 mg/L GGA standard solution with different salinity gradients (0–2.0 mol/L NaCl artificial seawater) were tested. As shown in Figure 3, with the increase of salinity, the signal response of the BFR sensor decreased gradually. It was reported that high salinity significantly affects physical and biochemical properties of activated sludge, which then impacts membrane permeability [28]. Therefore, high salinity can affect the respiratory activity of the immobilized microorganism. The degradation activity to organic matters was inhibited as the salinity increased, and the signal response of the BFR sensor decreased correspondingly. However, it still maintained enough activity in the studied range of salinity. The results indicated that it can be used for rapid BOD monitoring of seawater.

#### 3.1.4. Stability of the BFR Sensor

The stability of the BFR sensor is a very important parameter in the rapid BOD measurement, which directly affects its application in the practical water sample [29]. The acclimation of freshwater microorganisms with different saline gradients might affect the activity and service life of the immobilized microorganisms on the BFR inner surface. Figure 4 shows the output of the BFR sensor during the 60-day test, which was carried out continuously at 35 °C at the interval of every other day. The response signal was quite stable during the investigated period, indicating that the domesticated microorganisms on the BFR sensor can be reproduced by themselves. The stable performance is due to the flow-type structure design. After the water sample was injected into the reactor, the hydraulic shear force can maintain the biofilm stability and limit its overgrowth. The reproductive ability is crucial for the stability of the BFR sensor. The result implied that the BFR sensor can be used for the long-term BOD test.

### 3.2. Activity and Application of the BFR Sensor

#### 3.2.1. Degradation Ability for Different Organics

Natural mixed microorganisms often exhibit broad-spectrum degradation ability of different organics with the synergistic effect [19,20,21,22,23,24,25,30]. Therefore, different organics were selected to test the degradation spectrum of the BFR sensor. As shown in Figure 5, the signal responses of the BFR sensor to different kinds of organics displayed excellent linear relationship with the results of conventional BOD_5_ method. The outcome indicated that the BFR sensor can degrade different organics without selectivity due to the synergistic effect of the domesticated microorganisms on the BFR inner surface. This is essential for rapid BOD sensors because the selected biodegradability is more inclined to lead deviated results from traditional BOD_5_ methods due to the complicated composition in the actual water samples.

#### 3.2.2. Standard Curve of the BFR Sensor

For the application of the BFR sensor, the standard curve of the BFR sensor should be established at first. The GGA standard solution with a set of concentrations was tested in 0.40 mol/L NaCl artificial seawater. It can be seen from Figure 6 that there is a good linear relation between the current signal and the BOD value in the range of 1.0–10.0 mg/L, and the correlation coefficient of the response curve is as high as 0.9951. The detection limit is 0.30 mg/L according to three times of signal-to-noise ratio. The results displayed that the BFR sensor prepared by domesticating fresh water sources has a good degradation ability in saline samples. Therefore, it is suitable for rapid BOD monitoring in both seawater and fresh water.

#### 3.2.3. Application of BFR Sensor

In order to test the universality of the prepared BFR sensor, both freshwater and seawater samples in different areas were tested. Three seawater points (Xiachuan Island, Fenghuojiao, and Sunshine beach) and four freshwater points (Tiansha River, Library pond, Lotus pond, and Weiming pond) were selected as the test samples. At the same time, the accuracy and reliability of the determined results were compared with the traditional BOD_5_ method. It can be seen in Table 1, the results of BFR sensor are roughly similar to those of traditional BOD_5_ results, while the errors of the seawater are greater than those of freshwater samples. The results can be attributed to the high salinity in seawater compared to the optimized conditions of 0.4 M NaCl, which affected the degradation activity of microorganisms. Even so, the natural microorganisms after the salinity acclimation still show excellent degradation ability. It shows that the determination results of rapid BOD have high accuracy and reliability. At the same time, it is universal for the detection of water samples in different environmental water bodies, and the application fields are expanded obviously.

### 3.3. Analysis of Microbial Community Structure

The community structure changes of natural microbial sources in the pond water before and after domestication were detected by high-throughput sequencing of 16S rRNA gene [31]. As shown in Figure 7, the original microbial community structure in the pond water is W1, and the community structure after domestication is W2. Additionally, the characteristics of corresponding microorganisms were shown in Table 2. The predominant proportion of the community structures is *proteobacteria*, which is widespread in both fresh water and seawater. It was decreased from 71.41% to 42.01% after the domestication. It was reported that elevated salinity has inhibitory or toxic effects on the microbial activity, and the removal efficiency of organic carbon would be decreased correspondingly [17,18,28]. Therefore, it should be attributed to the inactivation of some microbial species during the domesticating with saline water gradually. At the same time, *actinobacteria* decreased to a certain extent. However, *chloroflexi*, *cyanobacteria*, and *planctomycetes* increased obviously after domestication, indicating that they were enriched and propagated after acclimation with the gradual increase of salinity gradient. Compared the cultured microorganism with seawater microbial species, which displayed high activity only in seawater, the mixed microorganism can exhibit high nonselective degradation activity and stability in both fresh water and seawater.

## 4. Conclusions

A novel method for the BFR preparation by acclimating natural microbial sources is proposed. The method can universally determine BOD of both fresh water and seawater, overcoming the limitation that the BOD sensor can only be used in one kind of water body due to poor environmental adaptability before. Environmental microorganisms from fresh water were acclimated with artificial seawater of gradually increased salinity gradients successively, and the survived microorganisms exhibited tenacious vitality even in a high-saline environment. The domestication improves the adaptability of the selected microorganisms. Moreover, the BFR sensor does not need special daily maintenance, nor does it need to inject buffer solution to maintain microbial activity due to the adaptation from domestication, which reduces the secondary phosphorus pollution to the environment and still maintains high biological activity. By analyzing the community structure of natural microbial sources before and after domestication, it can be seen that most microorganisms do not lose activity and can adapt well to different water environments. The BFR sensor method has a promising wide application in the near future in the field of coastal areas affected by tides.

## Data Availability

The data presented in this study are available on request from the corresponding author.

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
