# Peer review of "A Universal Biofilm Reactor Sensor for the Determination of Biochemical Oxygen Demand of Different Water Areas"

_molecules, 2022, doi:10.3390/molecules27155046_

Round 1

Reviewer 1 Report

The authors report a biofilm reactor sensor for the determination of biological oxygen demand (BOD) in water. A number of BOD sensors were also proposed by the same authors. Apparently the advantage of the reported sensor is that it can be applied in both fresh water and seawater. In my opinion, the novelty and originality of the sensor is not sufficiently explained in the manuscript.

Following are some other concerns:

1) A full list of reagents should be included in the Experimental section

2) The current response at 37ºC and at a flow rate of 3.5 mL/min in figure 2 significantly differs from the response shown in figures 1 and 3 (in the absence of NaCl). This questions the reliability of the results at different flow rates.

3) A comparison of the analytical performance of BOD sensors should be included.

Author Response

Following are some other concerns:

  • A full list of reagents should be included in the Experimental section.

Answer: Thanks very much! The following is the list of experimental reagents. We added it in the supporting information.

List of Experimental Reagents

NO.

Reagent

 formula

 factory

1

Glucose

C6H12O6

Macklin

2

Glutamate

C5H9NO4

Macklin

3

Sodium chloride

NaCl

Guangzhou Chemical Reagent

4

Magnesium chloride hexahydrate

 MgCl2·6H2O

Guangzhou Chemical Reagent

5

Anhydrous sodium sulfate

Na2SO4

Guangzhou Chemical Reagent

6

Anhydrous calcium chloride

CaCl2

Guangzhou Chemical Reagent

7

Potassium chloride

KCl

Guangzhou Chemical Reagent

8

Sodium bicarbonate

NaHCO3

Guangzhou Chemical Reagent

9

Sodium bromide

NaBr

Guangzhou Chemical Reagent

10

Leucine

 C6H13NO2

Macklin

11

Lysine

C6H14N2O2

Macklin

12

Malic acid

C4H6O5

Macklin

13

Sorbitol

C6H14O6

Macklin

14

N-butanol

C4H10O

Guangzhou Chemical Reagent

15

Fumaric acid

C4H4O4

Macklin

16

Ethyl acetate

C4H8O2

Guangzhou Chemical Reagent

17

Xylose

C5H10O5

Macklin

18

Galactose

C6H12O6

Macklin

19

Ferric chloride

FeCl3

Guangzhou Chemical Reagent

20

Magnesium sulphate

MgSO4

Guangzhou Chemical Reagent

21

Potassium dihydrogen Phosphate

KH2PO4

Guangzhou Chemical Reagent

22

Disodium hydrogen phosphate

Na2HPO4

Guangzhou Chemical Reagent

23

Ammonium chloride

NH4Cl

Guangzhou Chemical Reagent

  • The current response at 37ºC and at a flow rate of 3.5 mL/min in figure 2 significantly differs from the response shown in figures 1 and 3 (in the absence of NaCl). This questions the reliability of the results at different flow rates.

Answer: Thanks very much for your helpful advice! We are sorry that there was a mistake in the previous manuscripts. The test temperature in Figure 2 should be changed from 37 ℃ to 35 ℃. We repeated the experiments, and all the current signals were within the error range.

  • A comparison of the analytical performance of BOD sensors should be included.

Answer: Thanks very much for your helpful advice! Because we could not find related BOD sensor which can be used in both fresh water and seawater at the same time, we cannot compare their analytical performance of BOD sensors at present. We should compare them in the near future according to your helpful suggestion.

Reviewer 2 Report

The paper in overall fits to the aim and scope of Molecules Journal, however, there are some comments:

1. The introduction part seems too short. It is recommended to extend this section with more detailed description of current state in the field of the research

2. It is recommended to provide the scheme of the bench

3. Results (and each figure) require more detailed explanation

4. It is recommended to provide add discussion section to the paper

5. Conlusions are poor

Author Response

The paper in overall fits to the aim and scope of Molecules Journal, however, there are some comments:

  1. The introduction part seems too short. It is recommended to extend this section with more detailed description of current state in the field of the research.

Answer: Thanks very much for your helpful advice! We try to revise the introduction part according to your advice as follows.

It is reported that high salinity environment significantly influences the microbial communities. [16]. And elevated salinity has inhibitory or toxic effects on the microbial activity and population structure in bioreactors, and it would reduce organic carbon removal efficiency [17, 18].

The BFR system embraces many notable advantages, such as simplicity in device, convenience in operation, and minimal maintenance, make it promising for practical BOD real-time warning.

What is more, the online analyzer based on BFR sensor can continually operate over 30 days without human intervention and additional chemical reagent consumption. It can track the fluctuation of the biodegradable organic compound level timely and accurately [25].

  1. It is recommended to provide the scheme of the bench.

Answer: Thanks very much for your helpful advice! We draw a scheme of the bench as follow. The BFR sensor and the standard solution/sample were placed in a 35℃ constant temperature water bath. The fully aerated water sample passed through the DO probe directly and output a stable current value. Then the water sample passed through BFR and output a stable current value through the DO probe. The measurement procedure takes 20 min. The difference between the current values is the analysis signal.

Scheme A1 The bench of BFR biosensor for BOD determination.

  1. Results (and each figure) require more detailed explanation.

Answer: Thanks very much for your helpful advice! We added related explanation according to your advice as follows.

What is more, the longer reaction time can lead to more DO consumption in the reactor.

It was reported that high salinity significantly affects physical and biochemical properties of activated sludge, which then impacts upon membrane permeability [28]. Therefore, high salinity can affect the respiratory activity of the immobilized microorganism. The degradation activity to organic matters was inhibited as the salinity increased, and the signal response of the BFR sensor decreased correspondingly.

The stable performance is due to the flow-type structure design. After the water sample injected into the reactor, the hydraulic shear force can maintain the biofilm stability and limit its overgrowth.

This is essential for rapid BOD sensors, because the selected biodegradability is more inclined to lead deviated results from traditional BOD5 methods due to the complicated composition in the actual water samples.

The results can be attributed to the high salinity in seawater compared to the optimized conditions of 0.4 M NaCl, which affected the degradation activity of microorganisms. Even so......

  1. It is recommended to provide add discussion section to the paper.

Answer: Thanks very much for your helpful advice! We put the discussion close to the related results in the “3. Results and Discussion” part of the manuscript. It should be better if we can provide an addition discussion section in the near future.

  1. Conclusions are poor.

Answer: Thanks very much for your helpful advice! We added related conclusion as follow.

Environmental microorganisms from fresh water were acclimated with artificial seawater of gradually increased salinity gradients successively, and the survived microorganisms exhibited tenacious vitality even in high saline environment. The domestication......

Reviewer 3 Report

Page 2, Line 81:

The authors highly suggested adding a picture or photograph of the newly developed BFR.

Please add the information regarding the working volume.

Page 4, Line 149:

The authors stated that “As shown 147 in Figure 3, with the increase of salinity, the signal response of the BFR sensor decreases 148 gradually”. Please elaborate the reason behind this observation.

Page 9, Line 219:

Please provide reference/citation for the statement “… attributed to the inactivation of some microbial species during the domesticating with saline water gradually”.

Author Response

Page 2, Line 81:

The authors highly suggested adding a picture or photograph of the newly developed BFR.

Please add the information regarding the working volume.

Answer: Thanks for your helpful advice! The BFR sensor photos before and after the domestication were given in Scheme A1. There are 75 pores in the column, and the pore diameter is 0.25 cm, the length is 7.0 cm. Therefore, the pore volume is 0.343 cm3. And the total volume is 25.76 cm3. We added it in the experimental part.

Scheme A1 The photos of the BFR sensor. a is the original printed reactor, b is the cultured BFR sensor, c is the profile of the BFR sensor.

Page 4, Line 149:

The authors stated that “As shown 147 in Figure 3, with the increase of salinity, the signal response of the BFR sensor decreases 148 gradually”. Please elaborate the reason behind this observation.

Answer: Thanks very much for your helpful advice! It was reported that high salinity significantly affects physical and biochemical properties of activated sludge, which then impacts upon membrane permeability [E. Reid, X. Liu, S.J. Judd, Effect of high salinity on activated sludge characteristics and membrane permeability in an immersed membrane bioreactor, Journal of Membrane Science 2006, 283, 164-171.] Therefore, high salinity can affect the cytomembrane permeability of the immobilized microorganism on the BFR sensor, thus affecting the respiratory activity of the immobilized microorganism. The degradation activity to organic matters was inhibited as the salinity increased, and the signal response of the BFR sensor decreased correspondingly. We revised it in the manuscript.

Page 9, Line 219:

Please provide reference/citation for the statement “… attributed to the inactivation of some microbial species during the domesticating with saline water gradually”.

Answer: Thanks very much! The high salinity environment significantly influences the microbial communities. [Wang, X. H.; Chen, Y.; Yuan, B.; Li, X. F.; Ren, Y. P. Impacts of sludge retention time on sludge characteristics and membrane fouling in a submerged osmotic membrane bioreactor. Bioresource Technology 2014, 161, 340-347]. And elevated salinity has inhibitory or toxic effects on the microbial activity and population structure in bioreactors, and it would reduce organic carbon removal efficiency [Frank, V. B.; Regnery, J.; Chan, K.E.; Ramey, D.F.; Spear, J.R.; Cath, T.Y. Co-treatment of residential and oil and gas production wastewater with a hybrid sequencing batch reactor-membrane bioreactor process. Journal of Water Process Engineering 2017, 17, 82-94; Mutlu, B. K.; Ozgun, H.; Ersahin, M. E.; Kaya, R.; Eliduzgun, S.; Altinbas, M.; Kinaci, C.; Koyuncu, I. Impact of salinity on the population dynamics of microorganisms in a membrane bioreactor treating produced water. Science of the Total Environment 2019, 646, 1080-1089]. From the analysis of microbial community, we can conclude that some microbial species decrease after the domestication, which should be attributed to the gradual inactivation in the domestication process with high salinity. We revised it as follow.

It was decreased from 71.41% to 42.01% after the domestication. Previous reports verified that elevated salinity has inhibitory or toxic effects on the microbial activity and population structure in bioreactors, and it would reduce organic carbon removal efficiency [28-29]. Therefore, it should be attributed to the inactivation of some microbial species during the domesticating with saline water gradually.

Round 2

Reviewer 1 Report

The authors have improved the manuscript. The table in the supplementary materials section should be mentioned in the main text. Editing of English is still required

Author Response

Thanks very much! We added the list of experimental reagents in the supporting information and mentioned it revised manuscript according to your helpful suggestion. We also improved the language of the manuscript at the same time.

Reviewer 2 Report

Dear authors,

after the revision your manuscript looks much better, however, according to my previous comment #2 it was recommended to include a figure of the  bench into revised version of the paper. However, it was only included into cover letter but not into a paper

Author Response

Thanks very much for your helpful advice! We put the bench in the revised manuscript according to your helpful suggestion in page 5 as follow.
